# Use and Significance of Nursing Diagnosis in Hospital Emergencies: A Phenomenological Approach

**DOI:** 10.3390/ijerph18189786

**Published:** 2021-09-17

**Authors:** Jose Miguel Cachón-Pérez, Purificación Gonzalez-Villanueva, Marta Rodriguez-Garcia, Oscar Oliva-Fernandez, Esther Garcia-Garcia, Juan Carlos Fernandez-Gonzalo

**Affiliations:** 1Department of Nursing, European University, 28670 Madrid, Spain; marta.rodriguez@universidadeuropea.es (M.R.-G.); oscar.oliva@universidadeuropea.es (O.O.-F.); ESTHER.GARCIA@universidadeuropea.es (E.G.-G.); 2Department of Nursing and Physiotherapy, Alcala University, 28801 Alcala de Henares, Spain; purigonzalez57@gmail.com (P.G.-V.); JUANCARLOS.FERNANDEZ@universidadeuropea.es (J.C.F.-G.)

**Keywords:** nursing diagnostic/nursing diagnosis, emergency nursing, nursing process

## Abstract

*Background:* Professional nursing organizations recommend the use of nursing diagnosis to enhance and facilitate the standardization of care and the development of a common language used by nursing practitioners. In the clinical reality of hospital emergency departments, however, its use is controversial. The objectives of the research are (a) to explore the use of nursing diagnosis in hospital emergency departments, and (b) to describe the meaning of nursing diagnosis for hospital emergency nurses. *Methods:* A qualitative phenomenological study was conducted. A purposeful sampling and snowball technique were used. Data were collected using in-depth interviews, researchers’ field notes, and documental analysis. An inductive analysis based on Giorgi´s proposal was used to identify significant emerging themes from interviews and field notes. Seventeen participants with a mean age of 40 were recruited. *Results:* Three themes were identified. The results showed how the use of nursing diagnosis in hospital emergency departments depends on nurses to apply a working methodology in their practice, along with other dimensions such as the characteristics of emergency care, the type of health problems, and the complexity of care. *Conclusions:* The use of standardized language in emergency departments is complex due to the overcrowded nature of care in these settings.

## 1. Introduction

The nursing process and the use of nursing diagnosis in favor of a homogenization and standardization of the care we provide to patients and the use of a common and universal language used by nursing professionals are recommended by NANDA International and previous studies and other national associations in Spain, such as the Spanish Association of Nomenclature, Taxonomy, and Nursing Diagnosis (AENTDE) and the Spanish Society of Emergency Nursing (SEEUE) [1,2,3].

The nursing diagnosis is a clinical judgment that identifies a health problem and is the basis for the nurse to describe the objectives and interventions to be performed [4,5]. In Spain, in the field of emergency care, the SEEUE recommends that it is necessary to use the nursing process along with nursing diagnoses to maintain scientific, professional, and quality assurance [6] in order for a nursing language with a high degree of consensus and standardization to be achieved, thanks to the different international taxonomies and classifications which reflect the stages of the nursing practice process, understanding the nursing diagnosis and its language as a development opportunity for the nursing profession [5]. Nursing diagnosis in the emergency department facilitates the resolution of ethical and social conflicts by the nursing staff [7]. Nursing diagnoses allow to organize the work of nurses in the emergency services, structuring through nursing objectives and interventions. Emergency department nurses find nursing diagnoses to be a difficulty rather than a facilitating tool for nursing care. Diagnoses are a work methodology that may not be applicable to the emergency department. 

Nursing classification systems are used to report nursing practices and maintain a unified documentation language. Nursing interventions classification (NIC) and nursing outcomes classification (NOC) are among the recognized terminology systems [7].

The objectives of the present study were (a) to describe the experiences of hospital emergency nurses in the use of nursing diagnoses; and (b) to describe the meaning of nursing diagnoses in nurses working in hospital emergency departments.

## 2. Materials and Methods

The guidelines for conducting qualitative studies established by the Consolidated Criteria for Reporting Qualitative Research (COREQ) [8] and the Standards for Reporting Qualitative Research (SRQR) [9] were followed.

### 2.1. Design

The choice of design is determined by the research question [9,10]. In the present study, the question aimed at describing the participants’ experiences in the hospital emergency context and the use and meaning given to nursing diagnoses. A qualitative descriptive phenomenological study was applied.

### 2.2. Research Team

The research team was made up of six nurses, all of whom belong to the university environment (J.M.C.-P., P.G.-V., M.R.-G.) and three of whom share teaching and clinical activity (J.C.F.-G., E.G.G., O.O.-F.). Prior to the study, the positioning of the researchers was established in terms of their theoretical framework, beliefs, and their motivations for conducting this research Table 1 shows the readiness of the researchers on the subject to be investigated.

### 2.3. Setting and Participants

The emergency departments analyzed have an average of 30 emergency beds. The nursing professionals are responsible for the initial triage of patients and initial care until discharge or referral of patients to hospitalization units, as well as for the discharge or admission reports to the different hospital units.

A non-probabilistic sampling strategy was used in this study, with an initial strategy of purposive sampling for the recruitment of supervisors of the emergency unit, later continued with snowball sampling. It was the participants themselves who pointed out other potential informants as relevant informants due to their experience and knowledge [11].

Inclusion criteria were (a) nurses who were actively working in the emergency department, (b) with at least one year of experience working in hospital emergency departments, and (c) who voluntarily participated and signed the informed consent form.

We contacted the nursing directors in the hospitals, who were responsible for searching within the participating emergency departments for nurses interested in participating in our study. We obtained an initial list of 25 nurses, 18 of whom volunteered to participate in the study.

### 2.4. Data Collection

An unstructured in-depth interview was used, aimed at understanding the participants’ perspectives on the phenomenon under study. A question guide was used with the topics to be discussed, but not closed, to allow the flow of information between researcher and informant. Table 2 shows the questions used by the researchers to interview the participants.

Interviews were stopped when data redundancy was reached, that is, when no new information or relevant data were incorporated in successive interviews [12].

All interviews were recorded and field notes were made. The recordings of these interviews were transcribed verbatim. Subsequently, each informant was sent the transcript of his or her interview so that he or she had the opportunity to modify or eliminate any fragment of the interview or to make any clarification that the informant had in mind, without having to give any explanation or justification.

The interviews were conducted in the participating hospitals in a room attached to the emergency departments.

### 2.5. Data Analysis

Amadeus Giorgi’s proposal was applied. In the first stage/phase, the researcher identifies units of meaning and then groups these units into sub-themes and topics by common meanings, identifying main themes [13,14]. Two researchers with experience in qualitative studies performed the analysis of the interview data. First, an analysis of each interview was performed. Afterwards, the results of the initial analysis were subsequently merged in joint sessions, during which the data collection and analysis procedures were discussed. In the case of differences of opinion, theme identification was decided by consensus. No qualitative software was used on the data. 

### 2.6. Rigour

The quality criteria enunciated by Leininger were followed, such as credibility, confirmability, context significance, recurrent patterns, data saturation, and transferability [10,11]. In order to guarantee methodological rigor, information triangulation was used, both with the participants and with the literature found in reference to the phenomenon studied.

### 2.7. Ethical Considerations

The research has followed the propositions contained in the Helsinki declaration in reference to the ethical principles for medical research involving human subjects [15]. The study was approved by the Research Ethics Committee of the Hospital U. de Getafe (Code 28062017119912), Hospital U. Fundación Alcorcon (Code 11.13), and the Universidad European de Madrid (Code CIPI/21/040).

The participants’ data were processed according to the criteria of Law 15/1999, of 13 December on the Protection of Personal Data, removing their personal data and assigning a pseudonym to each informant. On agreeing to participate in the study, the participant signed an informed consent document with explanations regarding the study. At any time, the informant could withdraw or refuse to participate in the study without having to provide any justification.

## 3. Results

A total of 18 nurses working in emergency departments were recruited, the mean age of the participants was 40 years (SD 8.2). See Table 3 for characteristics of the participants.

Three themes were obtained: (a) experience of difficulties for use of nursing diagnosis; (b) experience of opportunity—professional development?, and (c) dilemmas and conflicts—a labor or professional labyrinth?. The results of the study describe the difficulties experienced by nurses in emergency departments in using nursing diagnoses, the opportunity to work under a nursing methodology, and the dilemmas and conflicts involved in providing care in services saturated by the demand for care and the management of nursing taxonomy. See Table 4 for the themes identified.

### 3.1. Experience of Difficulties for Use of Nursing Diagnosis

This makes reference to those aspects that nurses report as a difficulty for the use of nursing diagnosis in the emergency department.

The nurses noted how in the emergency department, time takes on a highly relevant importance, care is on demand, without any planning, with multiple reasons for consultation with different prognoses that make it difficult to create a care plan using nursing diagnosis as a methodological tool.

DP4: *“Emergency times do not allow for this type of analysis. What we always try to do in the short term is to solve acute problems at first, then the rest is a posteriori”.*

DP2: *“Because it does not adapt to the work, nor to the rhythm of work in the emergency department. Diagnoses need a period of time, that is to say, a period of time”*.

Another of the difficulties identified is that nurses experience nursing diagnoses as something strange and difficult to understand, alien to their daily care activity. They perceive it negatively. It was even pointed out that nursing diagnosis is imported from another culture and they feel forced to implement it. 

DP12: *“Well, one of the first points is that it is cumbersome to the point of satiety, we can look for any word we want. Cumbersome, petulant and pedantic. Possibly because you want to escape, you don’t want to look for the point of union in the diagnosis of all life”*. 

DP5: *“All this is very American, the problem is how we have it implanted and how we have inherited it from another type of culture, more American style; if you get into NANDA, NIC, NOC you either handle it a lot or you are incapable of knowing what you are talking about… And in the ER”*.

A final difficulty is the use of nursing diagnosis as a working tool to apply care. Participants relate how using nursing diagnostics does not add value. They do not find a relationship between the actual work in an emergency department and the theorization of nursing diagnoses. 

DP6: *“To give names to things that no one understands and if we called things by their basic name, when you say it, identify what you mean, without having to go to a dictionary of diagnoses...”*

DP10: *“…using those diagnostic labels, the patient would not understand what we are saying”*.

### 3.2. Opportunity Experience. Professional Development?

This theme describes opportunities to improve the professional position of nurses in emergency departments by using and applying nursing diagnosis. It highlights what the nurses interviewed define as “patient assessment in practice”. 

DP10: *“...you can give them a nursing assessment, an emergency nursing assessment, always an emergency nursing assessment, the four things of risk and vital...”*

However, they also refer to positive meanings of the use of nursing diagnosis in emergency departments. It is striking that these accounts are expressed or described with verbs in the conditional tense; the participants describe them as possible meanings if they were able to use nursing diagnosis. They observe how they would be able to use their own professional language, which would help them to identify the specific care in these services and which they consider would contribute to the development of the professional figure as an emergency nurse.

DP4: *“It would be like establishing a...Or a language or whatever of communication, which in the end might be the nursing diagnosis, but which would give real continuity to what we do”.*

DP11: *“To give it a name, I think it could be to give it a name. If we have considered it, giving it a name with a diagnosis”*.

DP1: *“I think it could help us to organize ourselves mentally, it would help us not to forget things, we would have a similar way of working, we would all be more unified”.*

### 3.3. Dilemmas and Conflicts. A Labor or Professional Labyrinth?

Emergency nurses are often faced with dilemmas in relation to the use of nursing diagnoses. On the one hand, they receive input on what is correct to do, using nursing diagnoses defined with recognized taxonomies (corresponding to what they are told they have to do in their daily practice). On the other hand, the professionals describe how this diagnosis uses a somewhat sumptuous terminology, which requires a laborious process in practice, which labels the patient and does not give them a sense of professional development. What the theory says is far removed from their usual practice. They are at a crossroads between what they are told is the right thing to do, what they believe they could do, and what they actually do on a daily basis.

DP11: *“So far NANDA and all its diagnoses are so general that they cannot be applied to emergency care. At least today, it is so theoretical and so... that it cannot be applied to these services(…)I know what happens is that we put everything very bombastic”.*

DP1: *“Right now, in the day-to-day running of this emergency department, someone is talking about nursing diagnoses and we would say, ‘What are you telling me?”*

DP11: *“I would say theoretical, there are many that are not even understood, I read some and I say, but what is this”.*

DP7: *“We don’t use it among ourselves, neither to change the report, nor to count the patients on the floor. Look, we are now talking about nursing diagnoses and I see it as something, I don’t know it is not real, it is theoretical”.*

However, they spoke of how they perform activities called “emergency care”, which could correspond to what the theory states or dictates, but they do it in an undefined, intuitive way, without being truly aware of how they are doing it. 

DP: *“We are using it, but we are not naming it or saying let’s see what diagnosis does not come out”.*

DP8: *“Yes, it is done, but then when it comes to writing it down, it is not done in a standardized way”.*

DP11: *“We make many nursing assessments that we are not aware of, we are making nursing assessments and diagnoses to patients, what happens is that we don’t, we don’t have it in writing, it’s a little bit like that, isn’t it?”*

DP2: *“You make a mental diagram of what is happening to the patient or what could happen to him/her”*.

## 4. Discussion

Our results reflect how emergency nurses relate to the use of nursing diagnoses during their work in the emergency department.

Experience of difficulties for use of nursing diagnosis. We can observe how Pourhosseini et al. [16] describe the emergency department as one of the units that receive the greatest care pressure, based on the high occupancy and the structural and functional characteristics of the service. This could condition the application of methodologies for the development and application of nursing care in this context.

Previous studies show how the application of standardized language in nursing and the enunciation of nursing diagnoses present difficulties in the nurses’ perception of them. This occurs because nurses believe that this language does not adequately represent clinical practice or the health status of patients [17].

In addition, Rifà Ros et al. [18] pointed out that the terminology of nursing diagnoses is not the true language of nurses when talking about care. The same authors [18] reported that they did not capture the true essence of nursing work and are difficult to use in all contexts and may even become irrelevant. Rifà Ros et al. [18] showed that the terminology of these diagnoses creates difficulties for nurses to communicate effectively with each other, with other professionals and with patients and families.

Experience of difficulties for use of nursing diagnosis. Rivera et al. [19], in their review, call nursing diagnoses and the taxonomy used ‘academic language’, characterized by abstract, impractical, ineffective, time-consuming terminology that offers little benefit to patient care. It delves into how taxonomies are static tools and the static hardly brings advances in knowledge.

Gonzalez and Monroy [4] and NANDA [20], in the different definitions they put forward about the nursing process and in the identification of the stages of this process, always have nursing assessment first; as an essential step to give continuity to the following stages of the nursing care process. According to the results of the research, emergency nurses express a series of positive traits or distinctions in reference to the use of nursing diagnosis; specifically, if they had the opportunity to use it.

Brito [21] described how the nursing discipline, from the university, tries to strengthen and gain autonomy with its own body of knowledge and the use of clinical judgment. However, Brito [21] reported how there has been a problem with the practical application of nursing theories. The nursing care process was the ideal formula to carry out the objective, but the application of this process has not been viable in the face of the vortex and the excessive daily workload of the nursing units, and where sometimes nurses continue to work applying techniques.

Castner [22] described a series of difficulties in working with nursing diagnosis in the ED, highlighting the special characteristics of this type of service where nurses develop their care activity: different nursing actions and interventions, collaborative or not, that occur in short periods of time, and high responsibility decisions that limit the development and use of nursing diagnosis in the ED. However, it was proposed as an alternative to investigate and develop a specific work system for the ED, based on a methodology of nursing thinking, since it does show the need for standardization both in language and in nursing procedures in these EDs.

Dilemmas and conflicts—a labor or professional labyrinth? The nurses in their narratives speak of carrying out actions and activities in their usual practice, under the protection of a nursing care process, including some use of diagnosis. However, they express that this is done in a non-formal way; they refer to doing it in a non-defined or standardized way. Analyzing the aspect of clinical expertise and experience of nurses, Taghavi Larijani and Saatchi [23] showed that these condition their diagnostic capacity. She indicates that novice nurses focus especially on what they are able to observe, on clinical manifestations, signs, and symptoms; they are more comfortable performing the different stages of the diagnostic process until they reach the identification of a label. On the other hand, competent nurses are more able to grasp the globality of patients’ situations, incorporating patterns of knowledge and previous experiences that they synthesize until obtaining the possible identification of the problem, from a more precise clinical judgment.

Matney et al. [24] suggests a model where the expert nurse develops after having lived considerable experiences, which allow the professional to use intuition in decision making and does not depend on explicit knowledge. Specifically, Edwards [25] reported nursing science as a practical knowledge, a "know-how", and how environments are of vital importance and contribute to the daily learning of the professional in direct contact with the patient.

The authors of the present manuscript believed that the use of nursing diagnostics in the emergency department can serve to unify the way nursing professionals work and communicate, as well as an opportunity for professional development.

### Strengths and Limitations

Among the strengths of this study, to the best of our knowledge, this study described the experiences of emergency nurses regarding NANDA diagnoses and their use in clinical scenarios. Moreover, non-structured interviews and descriptions of researchers’ field notes were used to collect qualitative data. It is important to note that this study has certain constraints on the generalizability of findings, which limit the extrapolation of our results to the entire population of emergency nurses. Although the results cannot be extrapolated to the entire population, they can help nurse managers better understand the NANDA diagnoses use in emergency unit. 

## 5. Conclusions

Based on the results obtained, we observed that standardized nomenclatures and taxonomies such as NANDA, NIC, NOC are not the methodological tool used by nurses in hospital emergency departments; we found that it is a language that is difficult to use and is not adapted to the practice and casuistry of patients attending these hospital emergency departments. Regarding its applicability or implementation, we noted a series of conditions such as simplifying the definitions of diagnostic labels to improve the adaptability of this process to clinical practice; the possibility of dedicating more time to standardized records; and increasing training in standardized nomenclature. We emphasize as an alternative the use of a clinical language that is not specific to nursing professionals but is shared with other health professionals working in hospital emergency departments.

## Figures and Tables

**Table 1 ijerph-18-09786-t001:** The readiness of research.

Theoretical framework	Research is based on an interpretative framework, where the experience of a phenomenon may vary individually. There is no single, objectifiable reality; experience can modify the meaning given to a lived event or phenomenon.
Beliefs	Nursing diagnoses are a theoretical expression of nursing knowledge that, in some contexts, such as emergency services, can present difficulties in their application because they do not reflect the reality of the healthcare activity.
Motivation for the research	The aim is to give a voice to nurses in emergency departments in order to describe their experience with nursing diagnoses and NANDA language.

**Table 2 ijerph-18-09786-t002:** Unstructured in-depth interview guide.

Investigated Theme	Questions
**Nursing Diagnoses in the Emergency Department**	What is the nursing methodology used in the emergency department where you work?
**Overcrowding**	How many patients usually come to the emergency department on a typical day?
**Professional Development**	What are the professional development opportunities for nurses in the emergency department?
**Communication**	What communication system is used in your emergency department to interact with other healthcare professionals?

**Table 3 ijerph-18-09786-t003:** Characteristics of the participants.

Age	Media: 40
	SD: 8.2
Sex	Male: 4
	Female: 14
Time of Working	Media:19
	SD: 6.3

**Table 4 ijerph-18-09786-t004:** Themes identified.

Themes	Description
Experience of difficulties for use of nursing diagnosis	Nurses believe that nursing diagnoses implement the complexity of nursing care.
Experience of opportunity—professional development?	This topic refers to the possibility of improving the nursing profession through the use of nursing diagnostics.
Dilemmas and conflicts—a labor or professional labyrinth?	The discussion on working methodology is a boost to clinical care in the emergency department for nurses.

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
