# Peer review of "Use and Significance of Nursing Diagnosis in Hospital Emergencies: A Phenomenological Approach"

_ijerph, 2021, doi:10.3390/ijerph18189786_

Round 1
Reviewer 1 Report
Dear Authors,
I carefully read your work dealing with an interesting topic.
To improve your manuscript, I suggest some revision as follows. Thank you in advance for your consideration.
Abstract
From line 15 to line 19, there are some redundancies. Please check and revise.
Introduction
I think this section should better describes the reasons why this study was necessary.
To allow readers to understand the importance of the application of standardized terminologies in clinical practice it would be useful to report evidence related to the impact of nursing diagnosis on patients’ outcomes. After that, you should also describe literature gaps related to the application of nursing diagnoses in hospital emergency settings.
Methods
As regard participant and setting descriptions, some information could be added to improve clarity of your methods. Please describe how participants were approached and where data collection was performed. Have you used a software to manage the data?
Results
I suggest reporting a table in which themes and categories are clearly identified. Accordingly, revise (where necessary) the format of narrative results.
Are you sure that is correct to report the names of the interviewed nurses?
Discussion
I think discussion section should be oriented to discuss and link with literature your main results rather than predominantly report the results of literature. I noted that most of the work has been done. You can better highlight emerged themes and link them with references.
In addition, a brief description of strengthen and limitations of your study is expected.
Conclusion
I suggest to better point out that you described in results section.
Author Response
We would like to thank the Editors and the Reviewers for their careful consideration of our manuscript. We would also like to thank the Reviewers’ suggestions, which we believe have enhanced the quality of the manuscript. We have highlighted all the changes we have made throughout the text in yellow highlight. Below please find a detailed list of how we have addressed each comment.

Reviewer 2 Report
Thank you for reviewing this paper, you conducted a qualitative research related to nursing diagnosis in emergency department. Nursing diagnosis is very important to proceed nursing process, but it’s not easy in emergency setting. Therefore, this paper is meaningful, but I give the comments on each section below. I wish my comments can give developing your study.
Abstract:
- Some sentences are repeated. For example, ‘A phenomenological qualitative study (line 15)’, ‘A purposeful sample and snowball techniques were used. In depth interview was used, and documental analysis (line 17-19) was applied. Please deleted repeated sentences.
- Number (3) for Results is needed. Please insert ‘(3) Results’ in front of sentence with ‘The results~ (line 19)’
- Conclusion didn’t describe significant of this paper, because you wrote your second purpose of this study was ‘b) to describe the meaning of nursing diagnosis for hospital emergency nurses.’ So please add the meaning of the second result according to the second purpose.
Introduction:
- On Line 32, the SEEUE is repeated. Delete it.
- Once an abbreviation showed, the full name is not needed from the next. So, ‘the Spanish Society of Emergency Nursing’ and the round bracket should be deleted (line 35-36).
- If you want to use abbreviation, the full name is first. So, please write the full name of COREQ (line 52)
Materials and Methods
- 1 ‘Design (line 54) is typo. à Design
- Please, start with sentence. Change line 55 to sentence.
- Simple explanation of research method is not needed. Therefore, the second and third paragraph is not needed (line 59~68). They are better to be deleted.
- The title of <Table 1> is better to change. It is not researchers’ positioning, but readiness of research or something.
Results
- The first paragraph is better to show with table including information of all participants (line 126-129).
- For understanding of the qualitative results, please show the table with sub-theme.
- 1 title is better to change like “Experience of difficulties for use of nursing diagnosis (line 139)
- The narrative of participants is better to show italic letter in results part.
Discussion
- What is (ref)? (line 235)
- Discussion part should be showed your opinion not Matney’s comments. Especially, the last paragraph is more important to show with your final suggestion.
Author Response

(The authors gave the same response as above.)

Round 2
Reviewer 2 Report
Thank you for editing of authors as reviewer's comments.
I think the manuscript would be better.